# A C-Terminally Encoded Peptide, MeCEP6, Promotes Nitrate Uptake in Cassava Roots

**DOI:** 10.3390/plants14081264

**Published:** 2025-04-21

**Authors:** Fabao Lu, Xiuning Wang, Bo Liu, Hongxin Lin, Li Ai, Weitao Mai, Xiaochen Liu, Huaifang Zhang, Jinling Zhao, Luqman Khan, Wenquan Wang, Changying Zeng, Xin Chen

**Affiliations:** 1National Key Laboratory for Tropical Crop Breeding/Sanya Institute of Breeding and Multiplication/School of Tropical Agriculture and Forestry, Hainan University, Sanya 572000, China; lufabao1998@163.com (F.L.); wangxiuning313@163.com (X.W.); liubo090325@163.com (B.L.); aili19991108@163.com (L.A.); 18889164038@163.com (W.M.); liuxiaochen1680@163.com (X.L.); huaifangz@163.com (H.Z.); zhaojinling2024@163.com (J.Z.); luqmankhan@hainanu.edu.cn (L.K.); 994341@hainanu.edu.cn (W.W.); 2Institute of Tropical Bioscience and Biotechnology, Chinese Academy of Tropical Agricultural Science, Haikou 570100, China; 3Institute of Soil, Fertilizer, Resources and Environment, Jiangxi Academy of Agricultural Sciences, Nanchang 330000, China; lhxtfs@163.com; 4Sanya Research Institute, Chinese Academy of Tropical Agricultural Science, Sanya 572000, China

**Keywords:** C-terminally coding peptide, cassava, nitrogen use efficiency, nitrate uptake, growth hormone

## Abstract

Cassava, an essential food crop, is valued for its tolerance to infertile soils. This study explores the role of C-terminally encoded peptides (CEPs) in cassava, mainly focusing on *MeCEP6* and its function in nitrate uptake and plant growth. A comprehensive search on the UniProt website identified 12 *CEP* genes in cassava, predominantly located on chromosomes 12 and 13. Notably, *MeCEP6* demonstrated high expression levels in root tips and exhibited a significant response to low nitrate stress. Exogenous MeCEP6 and its overexpression enhanced NRT2 transporter expression while suppressing auxin-related genes, promoting nitrate uptake and inhibiting seedling growth under nitrogen limitation. This growth inhibition likely represents an adaptive mechanism, enhancing cassava’s survival under nitrogen limitation by optimizing nitrogen allocation and use efficiency, albeit at the cost of reduced growth potential in nitrogen-replete conditions. Moreover, it was identified that *MeWRKY65* and *MeWRKY70* could interact with the promoter of *MeCEP6* to modulate the expression of *MeCEP6*. The dual-luciferase assays further prove that *MeWRKY65* and *MeWRKY70* can activate the transcription of *MeCEP6* under low nitrate stress conditions. The study’s results help explain the underlying mechanism of *MeCEP6* that benefits nitrogen use efficiency and nitrogen deficiency tolerance in cassava. These findings provide a molecular basis for improving cassava yield in nitrogen-deficient soils and highlight *MeCEP6* as a potential target for crop improvement.

## 1. Introduction

Nitrogen is one of the plant macronutrients that plays a critical role in plant growth and development; however, its availability in the soil mainly changes over time and in various areas because of the changes in multiple conditions [1,2]. Therefore, to avoid the impacts of fluctuating nitrogen, plants (including cassava) have developed complex ways to monitor changes in environmental nitrogen, which trigger considerable biochemical adaptations in root architecture to obtain the maximum nutrients from the soil nitrogen [3]. Nitrate is an essential form of nitrogen used by plants directly as a nutrient and potentially as a signal molecule to modulate nitrogen-dependent gene expression and coordinate downstream adaptive responses to maintain a balance between plant growth and nutritional stress adaption [4,5,6]. Local and systemic long-distance signals regulate the root absorption and adaptive response according to the nitrogen status within and outside the plant [7]. It has been proved that proteins like NRT1.1 and NLP7 can directly sense nitrate levels and then receive and transduce nitrate status signals [8,9,10]. Some transporters in the NRT1 family have been capable of transporting nitrates along with other compounds, including nitrite, amino acids, peptides, and various plant hormones, suggesting that they may work together to regulate plant growth and development [11,12,13,14,15].

C-terminally encoded peptides (CEPs) are small peptide hormones produced by plants, including an N-terminal signal peptide, a connecting peptide, and one or more functional domains at the C-terminal area [16,17]. They are made up of approximately 20 amino acids. They have been conserved very well across species, regulating plant growth and development, stress response, and nutrient uptake [18]. CEPs bind to receptors on the cell membrane and activate the expression of downstream genes, thereby affecting plant physiological activities [19]. The overexpression of *AtCEP1* inhibits primary root elongation, reduces the number of lateral roots, and results in smaller cells in the apical meristem and maturation zone [20]. *CEPs* can function in several plant organs to regulate their development and further influence crop yield [21,22,23]. For instance, overexpressed *MtCEP1* reduces the number of lateral roots in alfalfa but does not affect root length [24]. The expression of *AtCEP2* minimizes the number of rosette leaves and delays flowering [25]. In maize, the expression of *ZmCEP1* causes shorter plant height and smaller seeds, leading to a lower plant yield [26].

A previous study has proved that CEPs can be sensitively produced by the roots upon low-nitrogen stress, transported long-distance through the vascular bundles, recognized by its receptor *CEPR* on the above-ground part, activate the production of *CEPD*, and induce the expression of the *NRT2.1* gene, enhancing nitrate absorption in the roots [27]. *HBI1* and *TCP20* act in concert to enhance *CEP* expression, improve nitrate absorption efficiency, and maintain nitrogen levels within *Arabidopsis* [28]. In addition, local applications of the CEP1 peptide can also promote nitrate absorption in plants such as *Arabidopsis* and alfalfa, thereby promoting their growth and development [6].

Cassava (*Manihot esculenta* Crantz) is the world’s sixth most important food crop, growing in more than 100 countries and regions in Africa, the Americas, and Asia. This plant performs well in infertile soil [29,30]. However, even with the advancements made in cassava research, the mechanisms behind its tolerance to poor soils are not yet fully understood, and the specific role of *CEPs* in cassava needs to be explored [31]. This study predicted 12 cassava *CEP* homologous genes, and *MeCEP6* was identified as the highest level in the root tip zone upon nitrogen starvation. The overexpression of *MeCEP6* and the external application of MeCEP6 can stimulate nitrate absorption by activating the nitrate transporter NRTs in roots and increasing the nitrate–nitrogen concentration of the plants. Moreover, the cultivation of *MeCEP6*-overexpresed transgenic cassava and the external application decreases the expression of auxin signal-associated genes, resulting in growth inhibition. This phenotype resembles the adaptive response to low nitrate stress, suggesting that MeCEP6 overexpression suppresses cassava growth while enhancing tolerance to nitrate limitation. These studies provide new insights and a theoretical basis for peptides, helping increase cassava’s nitrogen use efficiency under N deficiency.

## 2. Results

### 2.1. Identification and Analysis of the Cassava CEP Gene

To identify CEP genes in cassava, we searched UniProt and retrieved 12 CEP genes designated as MeCEP1-12. These genes were located on chromosomes 12, 13, 15, and 17 (Figure 1A). Structural analysis revealed that all MeCEP genes were composed of a single exon that did not contain introns. Except for MeCEP7, MeCEP10, and MeCEP12, the remaining MeCEP genes did not contain untranslated regions (Appendix A). A more detailed prediction of the structural domains of MeCEP proteins was carried out using InterProScan-5.25-64.0. The results indicate that all MeCEPs, except MeCEP2, were equipped with an N-terminal signal peptide.

Additionally, MeCEP3-6, MeCEP8-9, and MeCEP11 possessed a maturation domain. In contrast, MeCEP1-2 and MeCEP7 contained multiple maturation domains, whereas MeCEP12 lacked a maturation domain (Appendix A). The sequencing alignment of the 12 mature domain protein-generated sequence conservation Logo further revealed that MeCEP1-8 exhibited differences at amino acids 2, 3, and 7. Meanwhile, MeCEP10 and MeCEP12 had deletions at amino acids 10 and 11 (Appendix A).

Interestingly, a highly conserved helix structure was identified at the C-terminus among the 12 MeCEP proteins, consistent with the C-terminally encoded peptide (Appendix A). This helix structure spanned in the range of 89-322a, and the isoelectric point (pI) was in the range of 5–10 (Table A1). Subsequently, the CEP families of cassava and *Arabidopsis* were combined, and an evolutionary tree was constructed using MEGA6 (Figure 1B). The results indicate that *Arabidopsis* AtCEP1 is most closely related to cassava MeCEP6, exhibiting the highest degree of homology.

Gene duplication is of vital importance in the evolutionary process of species. It was confirmed by the collinear analysis of the CEPs gene family in cassava species. Excluding MeCEP10 and MeCEP12, four gene duplication events were detected in both species. These events were found to originate from post-speciation whole-genome duplications or segmental duplications. In cassava, MeCEP1 and MeCEP2 are corresponding duplicates of MeCEP8 and MeCEP7. Notably, there are no corresponding orthologs of them in *Arabidopsis*, which implies that these duplication events took place after the divergence of the two species. In addition, the MeCEP5 gene in cassava is orthologous to AtCEP3 in *Arabidopsis*. However, the lack of an orthologous MeCEP3 in *Arabidopsis* indicates that this gene might have newly emerged during evolution (Appendix A).

A promoter analysis was conducted on the 2000 bp region upstream of the start codon of each MeCEP gene (Appendix A). The results indicate various elements related to different hormone and stress responses in the MeCEPs’ promoter. These encompassed elements responsive to abscisic acid, gibberellin, auxin, salicylic acid, and those associated with drought, low temperature, defense mechanisms, meristem activity, and seed-specific responses. Such findings imply that MeCEPs might potentially play significant roles in the context of abiotic stress responses.

### 2.2. Expression Patterns of MeCEP Genes in Different Tissues and Nitrogen Treatments

To study the expression patterns of MeCEP genes in cassava under different tissues and nitrogen conditions, we analyzed their transcripts in 11 tissues [32]. The results show that most MeCEPs were highly expressed in root tips, with MeCEP6 having the highest expression (Figure 2A).

After pre-treatment with nitrogen-free (-N) or ammonium (NH_4_^+^) media, most MeCEP genes were upregulated by nitrate (NO_3_^−^), especially MeCEP6 and MeCEP8, which showed weaker responses to ammonium or -N induction. Conversely, after nitrate pre-treatment, most MeCEPs were induced under -N conditions, with MeCEP6 being the most sensitive to nitrogen deficiency (Figure 2B). Thus, MeCEP6 was selected for further study due to its strong response to nitrate and nitrogen deficiency. Additionally, in 15-day-old SC8 seedlings transferred to -N MS medium, MeCEP6 expression increased after 12 h and peaked at 24 h (Figure 2C). As external nitrate concentrations rose, MeCEP6 expression decreased (Figure 2D), indicating that it is induced under low nitrate conditions and inhibited under normal nitrate levels.

### 2.3. Exogenous Application of MeCEP6 Promotes Nitrate Uptake by Cassava Roots

To investigate the role of MeCEPs on nitrate uptake, we synthesized the mature MeCEP6 small peptide (GWMPDGSVPSPGVGH). Ten-day-old SC8 seedlings were transferred to nitrogen-free MS liquid medium supplemented with 0 mM NO_3_^−^, 1 µM MeCEP6 peptide, 5 mM NO_3_^−^, or a combination of 5 mM NO_3_^−^ and 1 µM MeCEP6 peptide for 15 days of cultivation (Figure 3A). The exogenous application of 1 µM MeCEP6 peptide alone did not significantly impact the growth of SC8 seedlings in terms of root length and fresh weight compared to the 0 mM NO_3_^−^ treatment. In contrast, the sole application of 5 mM NO_3_^−^ significantly promoted plant growth. However, when it was combined with the exogenous application of 1 µM MeCEP6 peptide, an inhibitory consequence on plant growth emerged, manifested as shortened root systems and diminished biomass (Figure 3B,C). SC8 seedlings showed similar shoot nitrate concentrations but lower root nitrate concentration between 1 µM MeCEP6 peptide-only treatment and 0 mM NO_3_^−^ controls. As expected, 5 mM NO_3_^−^ treatment increased shoot nitrate concentration 10-fold and whole-plant levels 5-fold compared to 0 mM NO_3_^−^ controls, respectively (Figure 3D–F). Notably, 1 µM MeCEP6 peptide enhanced nitrate utilization efficiency under 0 mM NO_3_^−^ conditions, though combining 5 mM NO_3_^−^ with the 1 µM MeCEP6 peptide only elevated shoot nitrate concentration (Figure 3D, *p* < 0.05) without affecting roots, whole plants (Figure 3E,F), nitrate accumulation, or utilization efficiency (Figure 3G,H). These results indicate that the exogenous application of MeCEP6 peptide could improve nitrate utilization efficiency by decreasing the nitrate concentration in the root of cassava under low nitrogen conditions, suggesting a potential role in enhancing the tolerance of cassava plants to low nitrogen environments.

The qRT-PCR results show (Figure 3I–O) that, despite the seedlings being in a nitrogen-rich environment (5 mM NO_3_^−^), the expression of nitrate transporter genes *MeNRT1.1*, *MeNRT1.5*, *MeNRT2.1*, *MeNRT3.1*, and assimilation genes *MeNIA1* and *MeNIR1* in the roots continued to be upregulated within 48 h, showing a similar increasing pattern. This result suggests that *MeCEP6* can enhance the ability of roots to acquire nitrate by activating the expression of nitrate uptake and transport-related genes and, at the same time, transmit nitrogen starvation signals, even in the presence of abundant external nitrogen.

### 2.4. Overexpression of MeCEP6 Promotes Nitrate Uptake by the Roots of Cassava

To further clarify the influence of *MeCEP6* on nitrate uptake and signaling pathways, SC8 transgenic plants harboring an empty vector (CK), with two lines exhibiting high levels of *MeCEP6* expression, namely *MeCEP6*-OE#1 and *MeCEP6*-OE#2 (Figure 4B), were cultivated for 60 days in 5 mM NO_3_^−^ solid MS medium, and samples were collected for analysis (Figure 4A). The results reveal that, in comparison to the SC8 plants, the root systems of the *MeCEP6*-OE transgenic plants were noticeably shortened and their biomass was diminished (Figure 4C,D).

Similar to the treatment with 5 mM NO_3_^−^ and 1 µM MeCEP6, the root and shoot nitrate concentration in *MeCEP6*-OE transgenic plants was higher than those in SC8 transgenic plants carrying an empty vector, but there was no significant difference in nitrate utilization efficiency and nitrate accumulation (Figure 4E–I). It indicates that the overexpression and exogenous application of MeCEP6 promote the absorption of nitrate by cassava seedlings. Moreover, genes related to nitrate transport and assimilation, such as *MeNRT1.1*, *MeNRT1.5*, *MeNRT2.1, MeNRT2.4*, *MeNRT3.1*, *MeNIA1*, and *MeNIR1*, were all discovered to be upregulated in the root systems of the *MeCEP6*-OE transgenic plants (Figure 4J–P). This further substantiates the impact of *MeCEP6* on nitrate uptake and nitrogen starvation signaling.

### 2.5. External Application of MeCEP6 and Its Overexpression Suppress the Gene Expression Associated with Plant Hormones That Regulate Growth

We observed that both the exogenous application of MeCEP6 and its overexpression notably reduced the length of the primary roots, diminished the fresh weight, and restrained the growth of the plants (Figure 3A and Figure 4A). To explore the influence of MeCEP6 on the root growth and development of cassava plants, we examined the expression of genes related to auxin synthesis and transport in the SC8 roots after a 30-day exogenous application of 1 μM MeCEP6 peptide. We also investigated the same in *MeCEP6*-OE#1 and *MeCEP6*-OE#2 plants following a comparable incubation period in the MS medium. The expression levels of three genes related to auxin synthesis, namely *TAA1*, *YUC2*, and *YUC4* (Figure 5A), along with four genes associated with auxin transport, including *AUX2*, *PIN1*, *PIN2,* and *PIN3* (Figure 5B), were all markedly reduced in the two transgenic plants when compared to those in the SC8 plants. The two *MeCEP6*-OE plants exhibited more potent inhibitory effects than the exogenous application. It can be inferred that *MeCEP6* can influence plant growth by impeding plant root elongation via the regulation of auxin biosynthesis and the distribution of auxin concentration among sub-tissues.

### 2.6. MeWRKY65 and MeWRKY70 Positively Regulate the Expression of MeCEP6

To explore the regulatory mechanism of MeCEP6, we employed the yeast one-hybrid (Y1H) assay and identified 11 potential candidate transcription factors (TFs; Table A3). Among them, two TFs, namely MeWRKY65 and MeWRKY70 (Figure 6A), exhibited relatively higher expression levels in the adventitious roots and root tips, respectively. In 10-day SC8 seedlings treated with 0 mM NO_3_^−^ and 5 mM NO_3_^−^ for 24 h, we found that the expression levels of MeWRKY65 and MeWRKY70 were increased under 0 mM NO_3_^−^ treatment compared with the 5 mM NO_3_^−^ treatment (Appendix A). We transiently transformed *Nicotiana benthamiana* leaves with Agrobacterium tumefaciens GV3101 (pSoup-p19) carrying the recombinant plasmids pGreen II 0800-MeCEP6pro together with pGreen II 62-SK-MeWRKY65 or pGreen II 62-SK-MeWRKY70. The results show that the LUC/REN ratios and fluorescence intensities from live-cell imaging were significantly higher in the combinations of pGreen II 0800-MeCEP6pro with pGreen II 62-SK-MeWRKY65 or pGreen II 62-SK-MeWRKY70 compared to the control groups. It indicates that MeWRKY65 and MeWRKY70 can bind to the MeCEP6 promoter and positively regulate its expression (Figure 6B–G).

## 3. Discussion

### 3.1. MeCEP6 Promotes Nitrate Uptake by Cassava Roots

Over 900 *CEPs* have been detected in plant genomes, and most of them respond to abiotic stresses, such as nitrogen, salt, and sugar [26]. However, the precise mechanisms through which *CEPs* govern nitrate absorption by cassava roots remain elusive. This study identified 12 *MeCEP* genes, 8 clustered on chromosomes 12 and 13. Most *MeCEP* genes are highly expressed in root tips, with *MeCEP6* showing the highest expression, particularly under nitrogen starvation, where *MeCEP6* expression is significantly upregulated. The exogenous application and overexpression of MeCEP6 upregulate genes associated with nitrate transporters and nitrogen assimilation. Furthermore, MeCEP6-treated plants show a higher NUE without significant changes in total nitrate accumulation or fresh weight under low-nitrogen conditions.

CEPDL2 can regulate nitrate absorption and transport through the stem-to-root migration [33]. Research shows that applying CEP peptides at 5 mM NO_3_^−^ promotes *MeGRXC1* (*MeCEPD*) expression in cassava leaves (Appendix A), indicating that CEP peptides may influence the absorption and utilization of nitrate in plants through a similar mechanism, improving nitrate use efficiency without changing total nitrate accumulation or fresh weight. The effects of CEP peptides on plants are time-dependent, involving initial adaptation, signal transduction, metabolic adjustments, and growth changes. In this study, with the CEP peptide treatment for 15 days, the nitrate utilization efficiency increased, but the total nitrate accumulation and biomass did not change significantly, which suggests that a prolonged treatment duration may be required to observe systemic effects on plant growth. In summary, *MeCEP6* enhances NUE under low nitrogen conditions by regulating nitrate transporter gene expression, thereby improving nitrate uptake in cassava roots.

### 3.2. MeCEP6 Inhibits the Growth of Cassava Plants Through Plant Hormones

In various crops, such as *Arabidopsis*, maize, rice, cereals, and apples, the expression of CEP peptide hormones has been demonstrated to impede plant growth and restrain root elongation [26,27,34,35,36]. In this study, overexpressed MeCEP6 or treated with exogenous MeCEP6 manifested shorter primary roots and diminished fresh weight. Exogenous MdCEP1 and its ectopic expression in *Arabidopsis* could negatively regulate genes related to IAA [36].

Similarly, *SiCEP3* impacts cereal growth by facilitating the uptake of abscisic acid (ABA) and curtailing root elongation [37]. *ZmCEP1* influences maize kernel development by modulating the transcription of genes implicated in nitrogen metabolism, nitrate, sugar transport, and IAA response pathways [26]. CEP signaling with cytokinin (CTK) impedes primary root growth via the CEPD system in *Arabidopsis* [38]. Moreover, investigations conducted by three other scientists highlighted that nitrate engages in crosstalk with other plant hormones like CTK and ethylene (ETH) [39,40,41].

While *MeCEP6* enhances nitrate uptake under low nitrogen, its concurrent suppression of auxin biosynthesis (*TAA1*, *YUC2*) and transport (*PIN1*, *PIN2*) genes (Figure 5) suggests a trade-off between nutrient acquisition and growth. It aligns with an adaptive strategy: cassava prioritizes nitrogen assimilation over biomass expansion, redirecting resources to sustain metabolic efficiency under nitrogen scarcity. Similar growth-inhibition phenotypes are observed in *Arabidopsis* and maize under CEP overexpression, supporting the hypothesis that CEPs act as systemic signals to balance nutrient uptake and growth under stress [27,42]. Based on these discoveries, we hypothesized that *MeCEP6* might impede cassava plant growth and development by holistically influencing IAA’s concentration and distribution. This inhibitory effect could also be achieved through synergistic crosstalk with other hormones like abscisic acid (ABA) and cytokinin (CTK).

### 3.3. The CEP Signaling Pathway Plays a Role in Cassava’s Tolerance to Barrenness

Cassava roots face heterogeneous nitrogen-stress environments. Low nitrogen in the rhizosphere triggers *MeCEP* gene expression in roots. MeCEP binds to MeCEPR in leaves, producing MeCEPD. MeCEPD moves to roots and activates genes related to nitrate transport and assimilation (e.g., *MeNRT1.1*, *MeNRT2.1*, etc.), enhancing root nitrate absorption capacity. So, cassava alleviates nitrogen starvation via the *MeCEP* pathway. Concurrently, *MeCEP* also suppresses genes related to growth-stimulating hormone synthesis and transport, leading to shorter primary roots and impeded growth. *MeCEP* likely affects plant growth and development by modulating the concentration and distribution of hormones, like IAA, CTK, ETH, and ABA, and substances like sugars [43,44]. Its role in cassava is complex, involving response to nitrogen stress and possibly regulating plant growth through hormone balance, and perhaps inhibiting growth under nitrogen restriction as a trade-off for barren tolerance. Future research needs to explore the interaction between *MeCEP* and plant hormones like IAA, ABA, and CTK in different environments, as well as the impact of *MeCEP6* on cassava growth and nitrogen use efficiency. It will help to comprehensively understand the role of the *MeCEP* signaling pathway in cassava’s tolerance to nutrient-poor conditions and its response to nitrogen availability.

## 4. Materials and Methods

### 4.1. Identification of MeCEP Genes in Cassava

Using the *AtCEP* amino acid sequence, we obtained 12 *MeCEP* genes in cassava from the search on UniPort (https://www.uniprot.org/; accessed on 8 November 2022). Evolutionary trees were generated using MEGA6. Mature peptides were predicted using InterPro (https://www.ebi.ac.uk/interpro/; accessed on 24 December 2022). We used the ExPASy proteomics server (http://expasy.org/; accessed on 15 May 2023) to predict isoelectric points and molecular weights—wepredicted the 3D model of the MeCEPs protein by I-TASSER (https://zhanglab.ccmb.med.umich.edu/ITASSER/; accessed on 3 July 2023). The cis-element in the *MeCEPs* gene promoter was predicted using PlantCARE (http://bioinformatics.psb.ugent.be/webtools/plantcare/html/; accessed on 8 August 2023).

### 4.2. The Creation of Overexpressed Transgenetic MeCEP6 Cassava

Primers corresponding to the full-length CDSs of *MeCEP6* were designed from the Phytozome v13 database. Total RNA was extracted from SC8 roots using the RNA Plant Extraction Kit (Tiangen Biotech, Beijing, China), followed by cDNA synthesis with PrimeScript™ RT Kit (Takara Bio, Shiga, Japan). The *MeCEP6* CDS (Manes.13G126400) was amplified using primers MeCEP6-F/R (Table A2) under the following conditions: initial denaturation at 98 °C for 30 s; 35 cycles of 98 °C for 10 s, 60 °C for 30 s, and 72 °C for 30 s, and a final extension at 72 °C for 5 min. The products were cloned into the pCAMBIA1300 vector via the KpnI and SalI restriction sites and transformed into Ecoli DH5α competent cells for propagation. Plasmid DNA was isolated and sequenced by Sangon Biotech (Shanghai, China https://www.sangon.com) using vector primers (MeCEP6OE-F/R, Table A2) to confirm sequence integrity. Next, the recombinant plasmid was transformed into the susceptible state of Agrobacterium LBA4404 and infected the brittle embryogenic callus of cassava SC8 [34]. After the transgenic seedlings grew new leaves, a real leaf was cut from the transgenic plants to extract DNA. Then, the detection primers were designed to identify positive transgenic events. For plants correctly detected at the DNA level, a one-month-old leaf sample was obtained and fully ground in liquid nitrogen for RNA extraction and reverse transcription. The qRT-PCR primers of the *MeCEP6* gene were designed to examine its expression level (Appendix A).

### 4.3. Plant Nitrogen Treatment

The cassava variety SC8 and two overexpression lines were grown on 1/2 MS solid medium (containing 10 mM NO_3_^−^ as the standard nitrogen level) for 10 days, and then transferred to a nitrogen-free (-N) MS liquid medium for stress tests. The control group for low-nitrogen experiments was strictly defined as 0 mM NO_3_^−^, while the standard nitrogen condition was maintained at 5 mM NO_3_^−^, unless otherwise specified. Potassium nitrate was the nitrogen source and potassium sulfate replaced it at lower levels to maintain ionic balance. The medium’s pH was 5.8, containing 25 g L^−1^ sucrose and 1 g L^−1^ agar. The chamber was kept at 26 °C, 50% humidity, with a 14 h/10 h light/dark cycle and 250 µmol m^−2^ s^−1^ light intensity. Seedlings were treated with 0 mM (-N), 0.5 mM, and 5 mM potassium nitrate and chloride for 0 h, 2 h, 12 h, 24 h, and 48 h. The harvested roots were frozen in liquid nitrogen and stored at −80 °C.

For the root nitrogen separation treatment, the root of the 15-day-old SC8 plant was carefully divided into two equal parts with a sterile scalpel after a 15-day pretreatment with no nitrogen (-N), 5 mM NH_4_^+^, or 5 mM NO_3_^−^, while keeping the stem system intact.After the no-nitrogen (-N) pretreatment, the roots of SC8 plants were treated with 5 mM NH_4_^+^ or 5 mM NO_3_^−^ simultaneously. For plants pretreated with 5 mM NH_4_^+^, the roots were exposed to either no nitrogen (-N) or 5 mM NO_3_^−^, whereas for those pretreated with 5 mM NO_3_^−^, the roots received 5 mM NH_4_^+^ or no nitrogen (-N). Samples were collected after 2 h and 2 d, quickly frozen in liquid nitrogen, and stored at −80 °C for subsequent RNA extraction and gene expression analysis.

MeCEP6 small peptide (AFRPTYPGHSPGVGH) was 75% pure and N-terminal-labeled with a fluorescent dye (5-FITC) and synthesized by Sangon Biotech (Shanghai, China, https://www.sangon.com). SC8 seedlings were cultured on 1/2 MS curing medium for 15 days and transferred to 1/2 MS liquid medium supplemented with 1 µM MeCEP6 peptide. The other experiments were conducted after 0 h, 12 h, 24 h, and 48 h.

For the exogenous application of the MeCEP6 treatment, 10-day-old SC8 seedlings were transferred to a liquid MS medium supplemented with 0 mM NO_3_^−^ + 1 µM MeCEP6 peptide [19], 5 mM NO_3_^−^, and 5 mM NO_3_^−^ + 1 µM MeCEP6 peptide without nitrogen for 15 days. Transgenic seedlings were propagated by excising apical buds (1 cm) and transferring them to a fresh MS medium for 60 days under sterile conditions. Roots, shoots, and apical meristems were collected separately from plants grown under standard nitrogen conditions for tissue-specific expression profiling. For the nitrogen treatment, 15-day-old SC8 cassava seedlings were pre-cultured in a 1/2 MS solid medium for 10 days and transferred to a liquid MS medium containing 1 µM MeCEP6 peptide for treatment. Treatment times were 0 h, 12 h, 24 h, and 48 h (3 biological replicates in each group). After treatment, root samples were frozen in liquid nitrogen and stored at −80 °C for subsequent RNA extraction and gene expression analysis.

### 4.4. RT-qPCR Analysis

RNA extraction was performed from cassava tissues (each 100 mg) using an RNA Plant Extraction Kit (Tiangen Biotechnology Ltd., Beijing, China), following the manufacturer’s instructions. The extracted RNA was then quantified using a NanoDrop spectrophotometer (Thermo Fisher Scientific, Waltham, MA, USA). Reverse transcription was conducted using PrimeScript^TM^ RT kit (Perfect Real Time) (Takara Bio, Shiga, Japan). The specificity of the gene-specific primers was analyzed using the melt curve analysis. The cassava actin gene *MeActin* (Manes.13G084300) served as the reference, with primer efficiency validated, and the 2^−ΔΔCT^ method was employed for quantification to calculate the expression of the target gene [45]. Three biological replicates were conducted for the experiment, and the forward and reverse primers used in the experiment are presented in Table A2.

### 4.5. Phenotypic Characterization

Representative plants exposed to the above nitrogen treatments were selected to measure the root length and weigh plant weight. We photographed and analyzed distinct phenotypes using image software, and then statistically evaluated the data to assess plant traits. The nitrate–nitrogen concentrations of the shoot and root were determined using a plant nitrate–nitrogen kit (Suzhou Kemin Biotechnology Co., Ltd., Suzhou, China, www.cominbio.com). The accumulation of nitrogen was calculated using the following formulas:Total NO_3_^−^ accumulation (mg) = Nitrogen concentration × Fresh weightNO_3_^−^ use efficiency (NUE, g/g) = Fresh weight/Total NO_3_^−^ accumulation

### 4.6. Yeast One-Hybrid (Y1H) Library Screening

The 2000 bp promoter region of *MeCEP6* was amplified with primers *MeCEP6pro*-F/R (Table A2), cloned into the pAbAi vector (Clontech), and then sequenced to verify this. A cassava root cDNA library was constructed using RNA from low nitrogen-treated seedlings. The yeast single hybridization (Y1H) system screened the cassava root cDNA library. After determining the minimum AbA concentration for screening, the cDNA library was transferred into the Y1H Gold yeast containing *MeCEP6pro*. Positive clones capable of activating reporter genes were screened on an appropriately selective medium. These positive clones were picked for PCR identification and Sanger sequencing after secondary streaking on SD/-Ura-Leu plates with minimum Aureobasidin A (AbA) concentration. The sequence data obtained from Sanger sequencing were aligned with the reference cassava genome for annotation. The function of candidate genes was further verified by co-expression and other functional verification experiments.

### 4.7. Luciferase In Vivo Imaging Assay (LCI)

The plasmid used in this experiment was pGreen II-0800, and the *MeCEP6* promoter was used as the reporter plasmid for this vector. Subsequently, effector plasmids were constructed by cloning the coding sequences (CDS) of *MeWRKY65* and *MeWRKY70* into the pGreen II-62-SK vector. The plasmids pGreen II-0800-MeCEP6 and pGreen II-62-SK-MeWRKY65 or pGreen II-62-SK-MeWRKY70 were co-transformed into Agrobacterium tumefaciens GV3101 (pSoup-P19). Then, they were cultured with shaking at 200 rpm in a LB liquid medium (containing 50 mg/L Rif and 50 mg/L Kan) at 28 °C until the optical density at 600 nm (OD_600_) reached 0.8. The bacterial suspension was injected into the leaves of Nicotiana benthamiana using the agroinfiltration method. Samples were collected after 48 h of dark incubation. The dual-luciferase activity (LUC/REN) was measured using the GloMax^®^ Navigator Multimode Detection System (Promega Corporation, Madison, WI, USA) according to the instructions of the Luciferase assay System kit (Promega Corporation, Madison, WI, USA). The fluorescence signals were captured by the ChemiDoc™ MP Imaging System (Bio-Rad Laboratories, Inc., Hercules, CA, USA) with an exposure time set to 30 s and a 530 nm emission filter.

### 4.8. Statistical Analysis

All experiments were conducted with three biological replicates, and the data presented in this paper represent the means ± standard deviation (SD). Statistical significance was determined using a one-way analysis of variance (ANOVA) followed by Tukey’s honestly significant difference (HSD) post hoc test for multiple comparisons. The difference in letters was statistically significant (*p* < 0.05). The statistical tests for each figure or table are detailed in the corresponding figure legends or Section 2.

## Figures and Tables

**Figure 1 plants-14-01264-f001:**
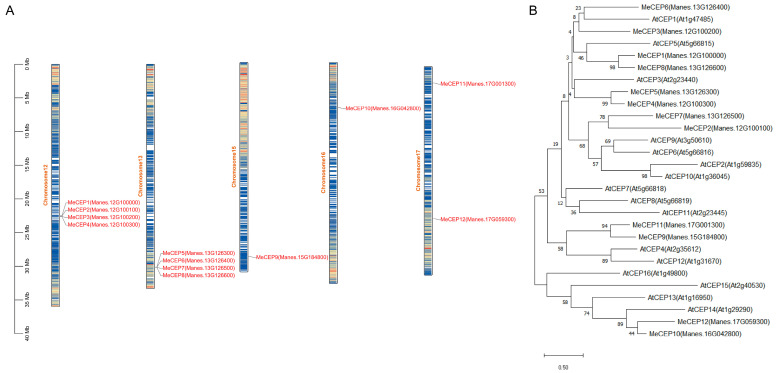
*MeCEP* gene family identification and phylogenetic tree analysis of Cassava and *Arabidopsis*. (**A**) Chromosome localization of the *MeCEP* gene family. (**B**) Evolutionary tree of *MeCEP* gene family and *AtCEP* gene family. Using the MEGA6 software, a phylogenetic tree based on amino acids was constructed using the neighbor-joining method.

**Figure 2 plants-14-01264-f002:**
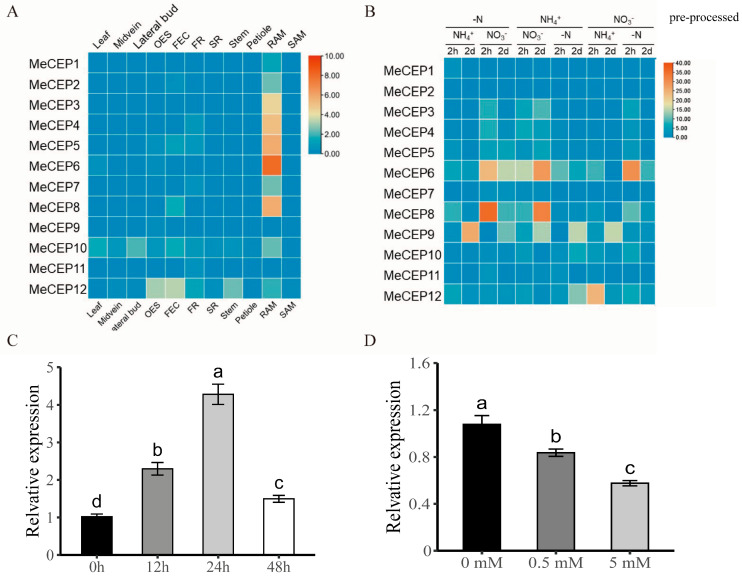
Expression profiles of the *MeCEP* gene family and their response to nitrogen treatments. (**A**) Heatmap of *MeCEP* expression across various tissues: mature endosperm (OES), friable embryogenic callus (FEC), fibrous root (FR), storage root (SR), root apical meristem (RAM), and shoot apical meristem (SAM). (**B**) Response of *MeCEPs* to split-root nitrogen treatments in 15-day-old SC8 plants pre-treated with -N, 5 mM NH_4_^+^, or 5 mM NO_3_^−^. (**C**) Relative expression of *MeCEP6* in roots of SC8 plants transferred from nitrogen-rich to nitrogen-depleted conditions over 0 h, 12 h, 24 h, and 48 h. (**D**) Relative expression of *MeCEP6* in roots of SC8 seedlings treated with 0 mM, 0.5 mM, or 5 mM nitrate for 24 h. Red represents a relatively higher level of gene expression, and blue represents a relatively lower level of gene expression, according to the value on the scale. The data represent the means of three biological replicates; different letters indicate significant differences (*p* < 0.05, one-way ANOVA).

**Figure 3 plants-14-01264-f003:**
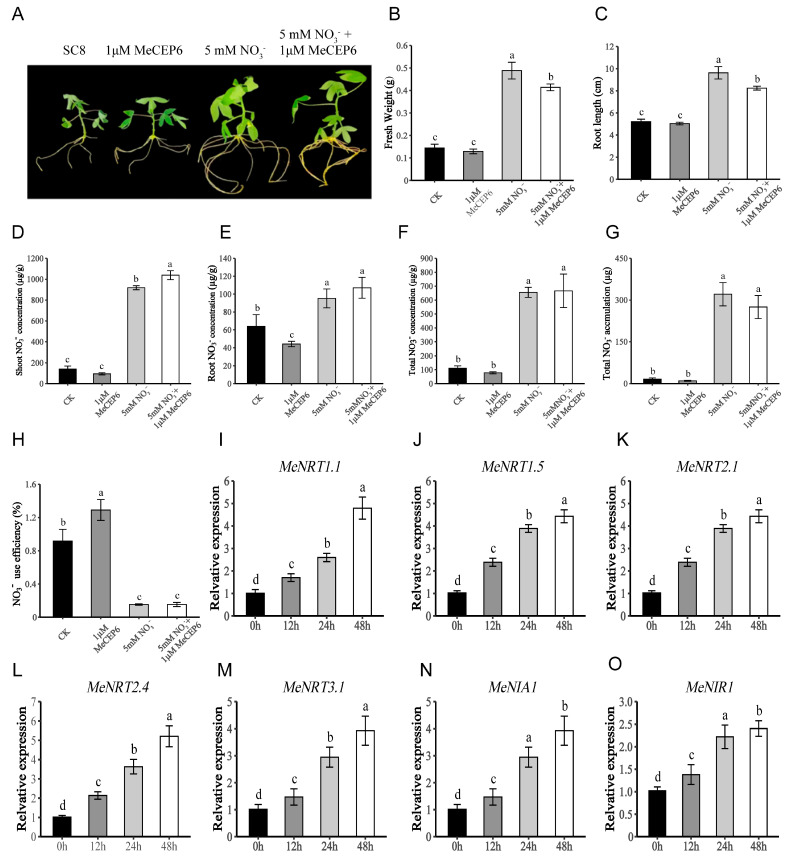
The effects of exogenous MeCEP6 peptide on cassava seedlings’ growth and nitrate absorption. (**A**) Phenotypes of SC8 seedlings treated with the control (CK), 1 µM MeCEP6 peptide treatment, 5 mM NO_3_^−^ treatment, and 5 mM NO_3_^−^ and 1 µM MeCEP6 combined treatment for 15 days. (**B**–**C**) The effects of different treatments on seedling fresh weight and taproot length, respectively. (**D**–**H**) The changes in nitrate concentration in the shoot, root, and total plant, and nitrogen use efficiency. (**I**–**O**) The relative expression levels of genes related to nitrate transport and assimilation in the roots after 1 day of 1 µM MeCEP6 peptide treatment. The experiments were repeated three times, with different letters indicating statistically significant differences (*p* < 0.05).

**Figure 4 plants-14-01264-f004:**
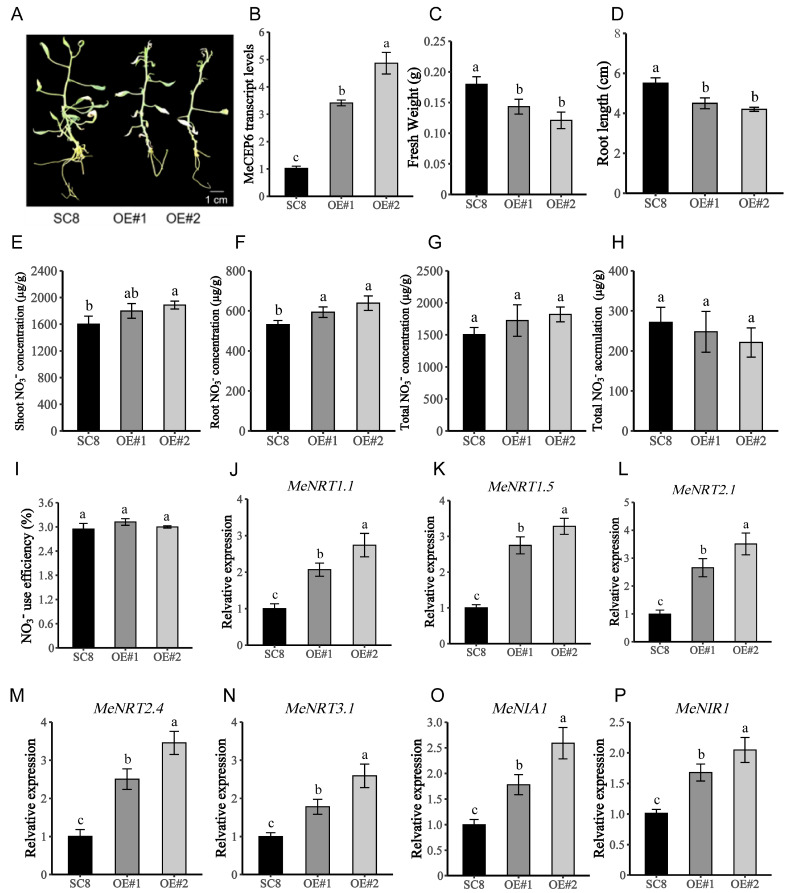
The effects of overexpression of *MeCEP6* on the growth and nitrate absorption of cassava seedlings. (**A**) Phenotypes of SC8, *MeCEP6*-OE#1, and *MeCEP6*-OE#2 after 60 days of culture under normal conditions. (**B**) Relative expression of *MeCEP6* in *MeCEP6*-OE#1 and *MeCEP6*-OE#2. (**C**,**D**) Overexpression of fresh weight and root length of transgenic plants. (**E**–**G**) Nitrate nitrogen concentration in shoot and root of transgenic positive plants, and total nitrate concentration in transgenic positive plants. (**H**,**I**) Nitrate accumulation and nitrate use efficiency of transgenic positive plants. (**J**–**P**) Relative expression levels of nitrate transport genes *MeNRT1.1*, *MeNRT1.5*, *MeNRT2.1*, *MeNRT2.4* and *MeNRT3.1*, and nitrate assimilation genes *MeNIA1* and *MeNIR1* in roots. Three biological replicates were performed for each trial, and significant differences were indicated by different letters at the *p* < 0.05 level. A one-way analysis of variance was used.

**Figure 5 plants-14-01264-f005:**
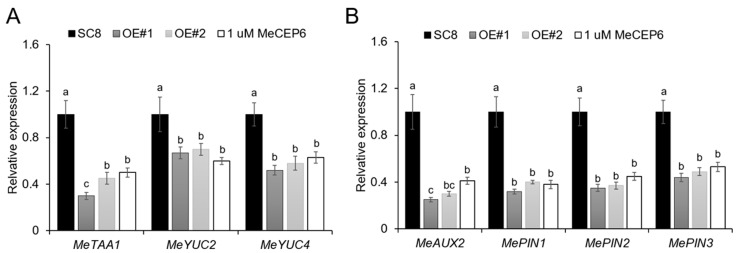
Exogenous and overexpression of MeCEP6 inhibited auxin-related gene expression. (**A**) Expression pattern of genes related to auxin synthesis in roots of SC8 plants, *MeCEP6*-OE #1 plants, and *MeCEP6*-OE #2 plants cultured in the MS medium for 30 days, SC8 plants treated with 1uM MeCEP6 peptide. (**B**) Expression pattern of auxin transport-related genes in roots. Three biological replicates were performed for each experiment, with different letters indicating significant differences between the data (*p* < 0.05), one-way analysis of variance.

**Figure 6 plants-14-01264-f006:**
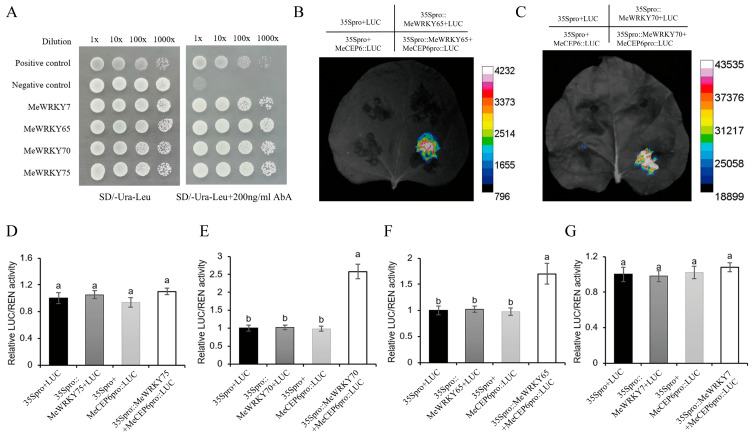
*MeWRKY65* and *MeWRKY70* positively regulate the expression of *MeCEP6* by directly binding to the promoter region of *MeCEP6*. (**A**) Interaction among *MeWRKY65*, *MeWRKY70,* and *MeCEP6*. This interaction in tobacco leaves was further validated by (**B**,**C**) dual luciferase assays. The LUC/REN activity alterations were visually presented by color transitions (shifting from red to blue), and the relative activity was precisely quantified with a bar chart. (**D**–**G**) β-galactosidase reporter assay demonstrated that these two transcription factors significantly increased the activity of reporter genes. Different letters indicated statistically significant differences between groups, with a significance level set at *p* < 0.05.

## Data Availability

The data supporting this study’s findings are available in this article’s Appendix A.

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
