# Peer review of "A C-Terminally Encoded Peptide, MeCEP6, Promotes Nitrate Uptake in Cassava Roots"

_plants, 2025, doi:10.3390/plants14081264_

Round 1

Reviewer 1 Report

Comments and Suggestions for Authors

You need to better define the N level of the control.  Is it truly no N or some other level.

What conditions were used for the different tissues searching the CEP expression in figure 2?  Was the tissue in Figure 2B roots or whole plants?  There is also no mention of this in the methods section.  This needs to be added.

Can you make a figure later on in the paper to show the different genes identified in the Y1H which bind the CEP6 promoter in response to the earlier expression data from figure 2.  This might show that the genes are expressed in the same tissues or a greater understanding of how the pathway is working.  It says it is in the suppl. files but I can not find this file.

I am a little lost on the mechanism.  So how does a gene indued by low N in the roots lead to less root growth.  Should not the opposite be true that the CEP should increase root growth under low N conditions?  I am not sure why an upregulation of auxin and N regulated genes is changing in response to a protein that inhibits growth.  

Comments on the Quality of English Language

Lines 53-56 is an example of the issues with clarity. 

Author Response

Please find the response enclosed in the attached pdf.

Reviewer 2 Report

Comments and Suggestions for Authors

The manuscript describes the properties of CEP genes in cassava, characterized by various methods. The work is meaningful. but interpretation of results is not always precise, and description of most methods are incomplete.

Issues in Results:

- Figure 2 A&B: color code is not fully resolved

- Most text in caption for Figure 2 is to be placed in Materials and methods

- Figure 3 D and E and corresponding description in the text: Nitrate content needs to be replaced by nitrate concentration of the tissues, according to the dimension indicated in the Figure.
Description of results in lines 185 – 188 conflicts with Figure 3, as according to the Figure only nitrate concentration of the shoot increased significantly by co-application CEP6.

- Legend for Figure 3 B-C gives reversed order

- line 210            probably “after 2 days”

- Figure 4 E and F and corresponding description in line 220-221: Nitrate content needs to be replaced by nitrate concentration of the tissues, according to the dimension indicated in Figure E/F. It may be emphasized that this value increased, while total amount of nitrate did not.

- lines 266-267
“MeWRKY65 and MeWRKY70, exhibited relatively higher expression levels in the adventitious roots and root tips, respectively”   Refer to supporting data, if appropriet.

Issues in Materials and methods:

This part is presented weakly, important descriptions are missing all along.

Section 4.2       cDNA is not extracted. Please give details of RNA isolation, cDNA synthesis, RT-PCR and cloning. Was the sequence verified? What were the sequences of “detection primers” ?

Section 4.3       How 15 days hydroponic cultivation was performed? Any changes of medium, aeration, etc?
I could not interpret the sentence: “The top buds of transgenic seedlings ..”. What does it refers to?

Section 4.4       Which actin gene of cassava served as internal control? How its use can be justified?

Section 4.6       This section has large gaps of information. For promoter cloning what primers were used, what sequence was amplified, and how was it cloned? Source/vendor of the vector?
Description of the yeast one hybrid screen is practically missing.
Sanger sequencing is a general practice, it says nothing about the platform or company that actually created sequence data.

Section 4.7       The same applies here, as for the previous section. Both need to be substantially more specific for all steps taken.
Some minor questions: “tobaccos of Ben” – what are they?
                                               “enzyme-labeled instrument” – what is it?
No words about image capturing, quantifying data etc.

Section 4.8       Please explain what “curing medium” is good for?
in lines 451-452 “other experiment” are mentioned. What are they?

Minor problems to be solved:

In some parts the text is incoherent, needs refinement:

line 27                 probably WRKY65 instead of WRKY60

line 41                 “biochemistry changes in root structure” makes no sense

line 53                 “small molecular weight? peptide”
                           the full sentence is best replaced, as it is not understandable as such

line  78-79         “and high-yielding plants” – is out of context

Fig 6 B,C,D,E       WAKY instead of WRKY

line 288               “UC/β-gal reporter gene system assay” – does not match previous descriptions

line 298               “These genes exhibit varied .. “ – please rephrase

Some typos found in the text need to be fixed
line
63          overexpressed
102        were equipped
124        orfan
261        to be deleted
469        WARKY or WRKY
486        theoretical
488        to be deleted

Ref #32 is incomplete

Author Response

Please refer to the annex for some of the contents.

Round 2

Reviewer 2 Report

Comments and Suggestions for Authors

The suggested changes and corrections have been made. The manuscript has been improved substantially.